# Universities in the National Innovation Systems: Emerging Innovation Landscapes in Asia-Pacific

**Venni V. Krishna**

School of Humanities and Languages, University of New South Wales, Sydney 2052, Australia;
v.krishna@unsw.edu.au; Tel.: +61-420557840

**Abstract:** Historically, universities and institutions of higher learning have gone through three academic revolutions, namely, teaching, research, and innovation. Universities and Higher Educational Institutions (HEIs) in the last two decades have come to occupy an important part in the national innovation systems (NIS), which is a complex of 'all important economic, social, political, organizational, institutional and other factors that influence the development, diffusion and use of innovations'. From a broader perspective, universities, together with public Research and Development (R&D) labs and science agencies, public policies (on industry, research, innovation and higher education, etc.) and business enterprises are now considered as important actors in the NIS of Asia-Pacific economies. The rise of Asia in the global knowledge-based economy from mid-1990s is closely associated with the rise of knowledge institutions of higher learning and scientific research output. Every Asia-Pacific country embraced and introduced policies relating to innovation in varying forms. Consultancy and collaborative links with industry being traditional forms of engagement, new policy and institutional measures in technology transfer and innovation to engage with society and business enterprises are gaining prominence. Policies for incubation, start-ups, and spin-offs, technology transfer offices (TTOs), and science and technology parks have gained tremendous prominence in leading Asia-Pacific universities. Different national innovation systems in the Asia-Pacific region have given rise to varying roles of universities. Whilst universities in Southeast Asian countries and India continue to play a traditional role of teaching and generating human capital, there are countries such as Singapore, China, Taiwan, and Japan, wherein universities are being transformed as entrepreneurial universities. Science and innovation policies in these countries have orchestrated the goal direction of universities as frontiers of innovation. Universities in Australia and New Zealand have so far been quite successful in marketing higher education to neighboring Asian countries. They have in recent years begun to embark on innovation and commercialization of research. The paper focuses on South East Asia and draws some comparison with more dynamic university ecosystems in East Asia. In doing so, the paper brings into focus the emerging innovation landscapes across the region.

**Keywords:** entrepreneurial universities; innovation landscapes; innovation ecosystem; innovation districts; Asia Pacific Universities

---

## 1. Introduction

Universities and HEIs in the last two decades have come to occupy an important part in national innovation systems (NIS) [1]. This is generally viewed as a complex of 'all important economic, social, political, organizational, institutional and other factors that influence the development, diffusion and use of innovations' [2]. A decade earlier, Freeman [3] defined it as a 'network of institutions in the public and private sectors whose activities and interactions initiate, import and diffuse new technologies'. From a broader perspective, universities, together with public R&D labs and science

agencies, public policies (on industry, research, innovation and higher education, etc.) and business enterprises are now considered as important actors in the Asia-Pacific economies. The rise of Asia in the global knowledge-based economy from mid-1990s is closely associated with the rise of knowledge institutions of higher learning and scientific research output. Universities play an important part not only in the nation-building process but also in meeting societal challenges whether it is in health, aging, sustainability, and climate change or in economic growth [4]. In the case of industrialized countries Mansfield found that one-tenth of the new products and processes commercialized from 1975 to 1985 could not have seen the face of market without substantial contribution from the academic research undertaken in universities [5]. Further, Grilliches drew attention to the fact that the rate of return on basic science (generally found in the academic research settings) is about three times that of applied R&D (generally undertaken in firms) [6]. In a most revealing way, Schapper, in his recent Chapter on universities and their role in economic development for a UNESCO Report on Asia pointed out, 'it is estimated that between 1988 and 2010, U.S. federal investment in genomic research generated an economic impact of $796 billion, while spending on the Human Genome Project between 1990 and 2003 amounted to $3.8 billion' [7].

Two features stand out that signify the transformation that is taking place. Firstly, there is the coupling of teaching and research for the advancement of knowledge, which indicates the research intensity in universities and HEIs. Secondly, the ability of these institutions to convert this research potential to have an impact on society and industry [8]. There are now some interesting studies to show the impact of universities in specific regions. There are classic cases of Stanford and MIT in boosting innovation ecosystems and development in Silicon Valley and Route 128 in Boston regions, respectively [9]. These developments came about over a long historical process involving broadly three overlapping phases of academic revolutions associated with three corresponding missions. After tracing the historical roots of these three academic revolutions, this paper explores the post-war experiences of some Asian economies. Having explored the historical role of universities, the paper attempts to capture how universities in the recent decades come to occupy a significant position in the respective national innovation systems. The rise of Asia in the 21st Century was possible, in a large measure, due to the part played by the research and innovation capacities of universities. The final section attempts to bring out the main innovation landscapes involving universities in the Asia-Pacific region.

## 2. Three Academic Revolutions

The first academic revolution came about when teaching in specialized higher educational institutions was institutionalized. In India, we can trace it to Nalanda and Takshashila, ancient universities in the Indian subcontinent. They functioned for more than 800 years between the 5th and 12th Century CE. Nalanda functioned as a residential university with 2000 teachers and 20000 students coming from India, China, Korea, Mongolia, Turkey, and Sri Lanka. It was the leading higher educational institution in the world in those times, which mainly focused on Buddhist studies, religion, culture, and civilization. The only other university that is reported in this era was Al-Azhar university in Cairo, established in 972 CE. At this time, it specialized in Islamic learning, including law along with logic, grammar, rhetoric, and how to calculate the lunar phases of the moon. Nalanda and Takshashila were destroyed around the 12th Century. The former has been re-established in its rejuvenated form under the former Chancellorship of Amartya Sen, N.L., since 2009–2010 [10]. Al-Azhar survives even today and emerged as a modern university to promote secular subjects as any other university in the world. The Pontifical and Royal University of Santo Tomas, the Catholic University of the Philippines was established as a private Roman Catholic university in 1611 [11]. In Italy, the first systematic evidence of institutionalized teaching is reported at the University of Bologna, founded in 1088; in France, at the University of Paris, founded in 1160 and at the universities of Oxford and Cambridge, which were established around the 12th Century. The first academic revolution phase continued until the beginning of the 19th Century.

The 'transformation of universities from institutions of cultural preservation', mainly teaching and maintenance of knowledge towards advancement of knowledge via research with a particular emphasis on science and technology disciplines, can be considered as the Second academic revolution [12]. Some scholars characterize this development as 'Humboldtian revolution'. For the first time, we witness the evolution of a model in the modern university that combines teaching with research. Martin Ben termed it as the 'Humboldt model', reflecting 'unity of teaching and research', which was successfully established at the Berlin University in 1810 by the Prussian educational reformer Wilhelm von Humboldt [13]. As Ben Martin goes on to argue, the essence of this model served as a model of the 'social contract' between state and society. The former, in the public interest, funds research in universities while maintaining a relatively high level of autonomy for both faculty and universities in their operation of teaching and advancement of knowledge. In the United States, teaching and research in the university systems came around the mid-19th century at institutions like Harvard and Colombia where professors, often inspired by their German doctoral mentors, sought to initiate research training and advanced degrees. This model spread quite rapidly since the 19th century to most parts of Europe, Asia, and North America with varying degree invoked by higher educational and science, technology, and innovation policies. For instance, consequent to the Vannevar Bush report of 1945 on Science: The Endless Frontier, the National Science Foundation (NSF) was established in the early 1950s. NSF became the major source of basic research funding for universities and standalone research laboratories.

The third academic revolution came about as universities progressed through further transformation taking on the mission of not just teaching and research but also getting involved in knowledge transfer and economic development. What is of significance is the development of coupling teaching and research with innovation and at the same time forging university and industrial links. As Etzkowitz, Webster, and Healey draw attention to, this phase came about more or less the same time as the second one [14]. Very well-known is the case of basic research in German universities that contributed to the German supremacy in the pharmaceutical and dyestuff industries. Much of the basic research and innovation potential in chemistry at the German universities found a way to feed into three major firms such as Bayer in 1863; Hoechst after 1880; and BASF in 1873. By 1877, Germany accounted for half of world's dyestuff production and captured the world market. In the years between 1908 and 1912, close university-industry relationships led to the process of synthesizing ammonia from nitrogen and hydrogen under high pressures, which came to be known as the Haber-Bosch process. All these firms have become big multinationals in the next 50 years. In the last decade and a half, perspectives on the 'triple helix' of university-industry-government relationships and the rise of entrepreneurial universities with specific cases of MIT and Stanford clearly signify the Third academic revolution [15–17].

The three phases in the progress and transformation of universities are not exclusive but overlapping missions. All these three missions and revolutions resonate, though in a small way, in the Asian–Pacific countries. Following the University of Melbourne in 1853 and University of Sydney in 1853, three Universities were established in the Presidencies of Calcutta, Madras, and Bombay as early as 1857. Before the close of the 19th Century, University of Tokyo (1867) and Keio University (1858) in Japan; Rangoon University (1878) in Myanmar; and Peking University (1898) and Tianjin University in China were established. The first two decades of the 20th century witnessed the creation of twelve other universities, namely, Tsinghua (1911) in China; Hokkaido University (1876), University of Tokyo (1877), Kyoto University (1897), Tohuku University (1907), and Kyushu University (1911) in Japan; Chulalongkon University (1917) in Thailand; Vietnam National University (1904) and Hanoi Medical University (1902) in Vietnam; Ewha Womens University (1910) and Yonsei University (1915) in South Korea; and the University of Hong Kong (1911).

The birth of modern universities in Asia from the mid-19th Century came about in different social, economic, and political contexts. The extent to which Asian universities experienced autonomy, and authority to impart quality education and institutionalize research and innovation activities

depended on respective political contexts. Most Asian countries were reeling under various colonial occupations from British, Spanish, Dutch, French, and Japanese control. Whereas China was under various dynasties until the Republic of China was established in 1949, Thailand was a Kingdom. Given this historical backdrop, different styles of university governance, curriculum, and modes of teaching and learning came to be institutionalized in modern Asian universities. Teaching was well institutionalized in all Asian universities, but the emergence of research and innovation activities greatly depended on university funding, generally governed by state policies. Unlike universities created under different colonial contexts, universities in Japan were relatively autonomous to progress into the second and third revolutions.

Soon after the Meiji Restoration in Japan, the University of Tokyo was established, followed by several imperial universities. The Japanese government made relentless efforts after the Meiji restoration to industrialize with the application of science and technology. The government had selected a few imperial universities such as Tokyo and Tohuku to develop engineering and technological capabilities by the First World War. Japanese investment into higher education and research in select universities was the single largest by the late 19th Century, particularly in communications and transportation [18,19]. One of the institutions established was the Kogakuryo Technical College in 1873, which later on became the engineering department of the University of Tokyo. Companies such as Toshiba, Dai Nippon Seiyaku in pharma, iron mills at Kamaishi and Yawata, and other leading iron and steel firms all received valuable technological inputs from the University of Tokyo in their formative years. The Iron and Steel Institute at the Tokyo University played an important part in university-industry relations and Japanese technological potential by the First World War [20,21]. Half a dozen imperial universities created in the 1870s and 1910 institutionalized research departments and institutions on university campuses. By the 1920s and 1930s, effective strong links and collaborations emerged between large firms and imperial universities [22]. Similarly, Tohoku University forged relations with industry with the establishment of the Institute for Materials Research (IMR) and the Research Institute of Electrical Communication (RIEC). Teaching and research were combined at Tohoku to evolve research and innovation potential for the development of the local region of Sendai. By the Second World War, research conducted at Tohoku had led to Japanese technological capabilities in material and electronic industries [23].

In India, the Presidency College was created in 1855 (earlier known as Hindoo College, established in 1817) became a center of industrial research with the pioneering work of Prafulla Chandra Ray, also known as the father of Indian chemistry. Ray's struggle against British colonial government for the lack of support for industrial research in 1893 led to the establishment of Bengal Chemical and Pharmaceutical Works (BCPW). In India, this was the first ever example of university-industry relationships. By the First World War, BCPW had become a major supplier of sulphuric acid and other chemicals to meet wartime needs and demands. It had employed 1400 workers by 1927 and emerged as India's leading pharmaceutical firms [24]. Ray's lead influenced other professors, such as T.K. Gajjar, who established a chemical plant around the same time as BCPW, which later grew into Alembic Chemical works Ltd. in Baroda in 1907. Gajjar was a Professor of chemistry at Baroda College and established a polytechnic institute known as Kala Bhavan [25]—the pre-cursor to the present-day M.S. University of Baroda. S.S. Bhatnagar, the first Chief of India's Council of Scientific and Industrial Research (CSIR), created in 1942, was heading the Chemical Laboratories at Punjab University in Lahore in 1940. His pioneering work on colloid chemistry, which earlier contributed to the promotion of oil industry in Assam, played an important role in the promotion of industrial research [24]. The third important example is the establishment of the Department of Chemical Technology at Bombay University in 1934, which played an important part in the establishment of National Chemical Laboratory of CSIR at Pune and chemical industrial cluster around Bombay and Pune since the 1940s [26].

### 3. Post-War Asian Experiences

Such early insights into university-industry relations, with the possible exception of Japan, should not be viewed as the emergence of second and third academic revolutions in Asian universities as a whole. By all means, these were isolated but important historical milestones. From the perspective of national innovation systems (NIS), the role of universities in most Asian countries was by and large confined to the expansion of HEIs and universities to produce human resources in all disciplines. The research intensity was confined to a small number of elite universities in different countries. One main reason for relegating university systems to mainly teaching and production of human resources was the science and technology policy, including higher education and research policies adopted in various Asian countries. It was not the Humboldtian model of developing teaching and research excellence at a national level assigning universities a prominent role in the NIS via science, technology, and innovation (STI) policies. The model that was followed in most Asian countries was somewhat closer to the French Model. This was characterized by the dominance of science and technology research, centralized and concentrated in specialized research laboratories such as French National Centre for Scientific Research (CNRS). This model also exemplifies a dual higher education system as a whole. It draws attention to the creation of elite HEIs, such as Grandes Ecoles, which attract the best of students and hence produce talents highly sought out in administration and civil services [27]. Even though these elite universities were not well funded or meant to carry out any substantial research, these institutions, by virtue of their national focus and standards of excellent teaching, instill a sense of superiority among students. This fits very well with the situation of Indian Institutes of Technology (IITs), Indian Institutes of Science (IISc), etc. in the Indian university system. Peking University and Tsinghua Universities, among a few others, in the Chinese case also fall under this segment of elite universities. The other tier of HEIs and universities are also not funded at par with elite universities but produce a mass of students contributing to greater higher education enrolment ratios [28].

Science and Technology policies in Asia led to the dominance of NIS by government-supported public research laboratories (PRL) under large science agencies. NIS models dominated by PRL such as Commonwealth Scientific and Industrial Research Organization (CSIRO) in Australia; dozen science agencies such as CSIR under the Ministry of Science and Technology in India; Korea Institute of Science and Technology (KIST), Korea Advance Institute of Science and Technology (KAIST), etc., under the Ministry of Science and technology in South Korea; and Science and Technology Agency, Japan, are examples from some Asian countries. In Thailand, Malaysia, Indonesia, Vietnam, the Philippines, Singapore, and other Asian countries, much of the national gross expenditure on research and development (GERD) is concentrated in such PRL and science agencies under government control. Throughout the post-independence era from 1947 to the 1990s, the large university system in India comprising over 500 universities and 20,000 colleges in 1998 drew less than 5% of national GERD. Over 68% of GERD is concentrated in government PRL and science agencies [29].

In almost all Asian countries, with the exceptions of Japan and South Korea, PRL and science agencies accounted for an average of 60% to 80% of GERD in the post-war era. In Japan, government PRL accounted for 31% of GERD in 1965, which came down to 20% in 1998. In South Korea, government PRL accounted for 53.5% in 1981 compared to 80% in 1970. This figure came down to 22% in the late 1990s. In both these countries, the share of business enterprise became a dominant source of national GERD from the 1980s and more prominently from 1990. STI policies in Korea were in a large measure steered by private business enterprises or government-supported chaebols. These enterprises were given a number of incentives to increase their share of GERD.

Universities in Korea, for a long time until about the 1990s, did not come to occupy an important position in the NIS. Lee observes, 'it is often been pointed out by industrialists that university graduates do not meet a standard that industries require due to theory-oriented education and research system in universities. University-industry R&D cooperation is, therefore, weak, being one of the problems of Korea's innovation system' [30]. The Japanese STI policy, which also assigned a greater role to business enterprises through Ministry of International Trade and Industry (MITI), demonstrates the

limited role played by Japanese universities. This situation prevailed for a long time in the post war-era until about the 1990s, when several reforms in universities and STI system came into force. As Odagiri points out, after World War II, a somewhat rigid system prevailed in the university system. Major universities being under the government control were restricted to access private funding [31]. Bureaucratic procedures did not foster industrial links in a large measure. They were not encouraged to apply for industry funds, apply for patents, and take on positions such as directors of a firm while occupying the position of a professor in a university.

In Taiwan, the Industrial Technology Research Institute (ITRI), the main actor in NIS, played a significant role through the 1980s and 1990s to develop endogenous technological capabilities in semiconductor technology. The government supported the ITRI in its efforts to develop most competitive technological capabilities through joint ventures and international collaboration. ITRI and the research innovation ecosystem developed by it could close the gap between the national and world frontier in semiconductor technology by the 1990s. ITRI was at the forefront of Taiwan's technological leadership in electronics and telecommunications. What was the role of higher education in this endeavor? Unlike the case of South Korea and, to some extent, Japan, 'universities played a very special role in East Asia development. Throughout this half–century universities were at the forefront in training generation after generation of highly skilled, technologically sophisticated graduates' [32].

In China, after founding of the People's Republic of China in 1949, much of the national GERD, as in other Asian countries, was concentrated in the government PRL and Chinese Academy of Sciences and Chinese Academy of Social Science laboratories or institutes. Lan and Zhou draw attention to two major reforms that are notable in the post-war era [33]. The first reform was in 1949–1955, when all universities, including several private universities, were brought under the fold of central government and consolidated on the Russian model. This involved leading and elite universities under the central government and others being administered by local or provincial governments. The second reform came about around 1985 after Deng's 1978 national reforms. This phase involved doing away with total control of central government through a process of decentralization, depoliticization, and infusing some form of diversity. The ideological focus of Cultural Revolution era was revised, and more autonomy was assigned to universities. From 1985, private universities were given permission, and by 2000, there were 1300 private universities in China. As Wu draws attention to and implies, until about the 1980s, universities in China had not come to occupy a major part in China's NIS [34].

## 4. Asian Strategies since the 1980s: Role of HEIs

The post-war era, particularly from the mid-1980s, can be seen as a turning point for various Asian countries. It may be pointed out that the major role of universities in Asia in the post-war era was to supply human resources and skills to strengthen the industrial base. As we will see, the role of universities as a potential innovation source began robustly only in the 1990s in Asian countries. The process of rapid industrialization and developing niche sectors of economy governed several Asian countries. Japanese success in industrialization based on technology transfer from the industrialized west and evolution of unique endogenous scientific and technological capacities led scholars to conceptualize 'late industrialization' and 'catching up' strategies [35,36]. What has come to be known as 'imitation to innovation' was the main title of Linsu Kim's volume on *The Dynamics of Korea's Technological Learning* (1997). Freeman was one of the first to draw attention to the way in which technology policies adopted by Japan and driven by its state industry ministry, MITI, led the country to enhance its economic performance, and from telecommunications to automobiles, consumer electronics and in 'megatronics', different policy instruments were adopted [3]. This included the approach of 'consortia' and subsidies to solve various problems. Together with high technology, even infant industries such as clothing, textiles, and machinery were given various incentives to develop technological innovation. Japan became an exemplar case not only for 'catch up' but provided an important reference to various East Asian countries such as South Korea, Taiwan, and Singapore. Success stories influenced various scholars and particularly the World Bank, which came out with

the publication *East Asian Miracle: Economic Growth and Public Policy* [37]. In contradistinction to market-related neoclassical explanations, there were important insights over the role played by state mediation and the importance of science, technology, and innovation policies [3,36]. This is, however, not the place to review and discuss all the factors and strategies. However, it is instructive to devote some space to understanding what role HEIs and universities played in the East Asian success story (South Korea, Taiwan, and Singapore) of 'catching up' and 'imitation to innovation'.

There is enough evidence to suggest that much of the East Asian success depended on the role of state mediation through science, technology, and innovation policies and big firms such as Toyota, Mitsubishi, Toshiba, Sony, and several others in Japan; LG, Samsung, Hyundai, and Daewoo in South Korea; and ASUS, Foxconn, Acer, and various other firms in Taiwan; and petrochemical, electronics, biomedical, and diagnostics and construction firms in Singapore. Even though the core knowledge and high technology did not emerge from HEIs and universities, they played a pivotal part in supplying high technology skills, training, and human resources in these large firms in different East Asian countries. As Richard Levin [38], President, Yale University, recently remarked, 'in the 1960s, 70s, and 80s, the higher education agenda in Asia's early developers—Japan, South Korea, and Taiwan—was first and foremost to increase the fraction of their populations provided with postsecondary education. Their initial focus was on expanding the number of institutions and their enrolments, and impressive results were achieved.' For instance, the number of university students per 10,000 people in Japan, South Korea, and Taiwan were much below USA and somewhat closer to that of Germany in the 1960s and 1970s. By mid-1970s, South Korea had overtaken Germany, and by mid-1990s, the USA. By 1990, Japan had overtaken Germany. Taiwan had overtaken Germany by 1980. In 1952, there were only four universities and four colleges with enrolment of 10,037 students in Taiwan. By 1989, they had expanded to 42 universities and 75 polytechnics, with a multi-fold increase of student enrolments [32].

In Korea, education at all levels, particularly higher education, expanded dramatically. By 1970, the school enrolment as a percentage of the population had reached 100% for elementary level, and 99% for middle school and 86% for high school by 1994. For tertiary level, it increased from a mere 3% in 1953 to around 20% in the 1970s, to register 49% in 1994. The total number of research scientists witnessed an increase of twelve times (1200%) from 10,300 in 1975 to 132,000 in 1996. In the case of PhDs, the number increased ten times (1000%) from 3417 in 1980 to 36,106 in 1996. Hence, the number of research scientists per 10,000 people increased from 4.8 to 29 for the same period [36].

A similar expansion of the human resource base can be seen in the case of Japan. First-degree graduates increased from 82,000 in 1975 to 130,000 in 1995; and master students increased from 10,000 in 1980 to 38,000 in 2000 in science, engineering, and agriculture sciences only. PhDs in these disciplines increased from less than 500 to nearly 10,000 for the same period. Between 1981 and 1999, Japan's R&D expenditure as a proportion of Gross Domestic Product (GDP) in HEIs surpassed those of the US, EU, and OECD from 0.34 to 0.45%. HEIs in Singapore in science, engineering, and management followed the same pattern of structuring higher education enrolments, keeping the demand in the high technology sectors. Singapore is also well known for attracting global talents. Migration policies are structured in a way to attract highly qualified skilled professionals for the emerging knowledge-based economy in leisure and tourism, biomedical, and knowledge-based devices in the Information and Communication Technologies, telecommunications, and gaming industry. Singapore was never known for its basic research base in any field up to 1990. The establishment of 'Biopolis' and 'Fusionopolis' with more than a dozen front-ranking research labs completely changed the city state's image as an important destination for scientists and highly skilled personnel [39].

When we investigate large developing countries such as India and China, these countries have expanded their HEIs and universities in all disciplines in the last few decades since the 1990s. However, unlike in Japan, in South Korea, Taiwan, and Singapore, the links between HEIs and industry were steered and restricted to certain niche sectors of economy. The well-known examples are ICT hardware, agriculture, and manufacturing in China and software, pharmaceuticals, and agriculture in India. For instance, since 1988, China has formed 52 high technology zones in ICT hardware at a close proximity

to HEIs and universities, the Zhongguancun Science Park in Beijing around the Peking and Tsinghua universities; and the Shanghai Zhangjiang high-tech and Zhuhai national high technology industrial zone in Shanghai. These account for over 70% of total industrial output [40]. Similarly, India's US $150 billion per annum software industrial sector was concentrated around ICT software technology parks in Bangalore, Chennai, Hyderabad, Noida-Delhi, and Pune between 2000 and 2015. These parks account for nearly 70% of total exports. The sector, as a whole, employed around 3.5 million software professionals drawn from Indian HEIs [41]. Similar structures of relevance between HEIs and emerging industrial sectors can be found in Malaysia's Multimedia Super Industrial Corridor near Kuala Lumpur; automobile component industrial sector in Thailand; and high technology electronic component sector in Vietnam. Whilst there is little doubt that highly skilled human resources and educational institutions, at all levels, played an important part in the East Asian success, no less important was the dramatic rise of knowledge output and production across Asian and pacific countries such as Australia and New Zealand. In other Association of South East Asian Nations (ASEAN) countries, universities and higher educational institutions played the role of supplying human resources to industry.

## 5. Higher Education: Growth and Enrolments

According to a recent UNESCO report, higher education enrolments in the world at large witnessed tremendous growth from 32.6 million in 1970 to 182.2 million in 2011. Little less than half of this (46%) expansion is accounted by East and South Asia [42]. As Table 1 shows, higher education, particularly gross enrolment ratios for bachelor's level students witnessed tremendous growth between 1990 and 2011, particularly in China, India, Malaysia, and Thailand. In some of these countries, enrolments increased by three to six times between 1990 and 2011. In most Asian countries, the role of private higher education has been an important factor in the expansion of higher education. In some countries such as China, Malaysia, Vietnam, and Cambodia, private institutions were not permitted. But after the 1990s, all these countries opened higher education to private players. In the last fifteen years, the private higher education has been the most dynamic growing sector in Asia than in other regions of the world. In 2011, over 40% of all students in higher education were enrolled in private institutions. In Japan, South Korea, Singapore, the Philippines, and Indonesia the figure ranges from 62 to 81 percent [42]. Expansion of higher education is also associated with rise in the growth of public expenditure on higher education. As Table 1 also shows, with the exception of Philippines and Thailand, most Asian countries increased their public higher education expenditures as a proportion of total publication education expenditure from 3 to 13 per cent.

**Table 1.** Higher education: gross enrolment ratio and expenditure.

| Country | GER for Bachelor Level Students | | GER in Higher Education | Public Expenditure on Higher Education as % of Total Public Education Expenditure | | Public Expenditure on Higher Education as% of GERD |
|---|---|---|---|---|---|---|
| | **1990** | **2011** | **2015** | **2000** | **2011** | **2015** |
| China | 2 | 11 | 42 | na | Na | 8 |
| India | 5 | 15 | 25 | 20 | 33 | 4 |
| Japan | 24 | 43 | 62 | 15 | 20 | 12 |
| South Korea | 28 | 69 | 87 | 15 | 18 | 9 |
| Malaysia | 3 | 19 | na | 32 | 35 | 29 |
| Philippines | 20 | 25 | na | 14 | 12 (-) | na |
| Thailand | 10 | 38 | na | 20 | 13 (-) | 20 |
| Indonesia | Na | 19 | 23 | Na | 20 | 34 |
| Vietnam | Na | 18 (2008) | na | Na | 15 | 5 |
| Singapore | Na | Na | na | 20 | 37 | 28 |

Sources: UNESCO (2014); World Bank 2017–2018.

## 6. Knowledge Production: 1990–2014

The rise of Asia in the global knowledge-based economy in the last decade and half from the mid-1990s is closely associated with the rise of knowledge institutions of higher learning and scientific research output. Even in the midst of the economic downturn of 2008, public policies of the leading economies of Asia continued to assign a very high priority to promoting institutions of higher learning and research. Together with a steady supply of highly skilled human resources, scientific output witnessed a dramatic growth during the decade and half from the mid-1990s. As shown in Table 2, economies such as Australia, India, and Japan increased their science output from 65% to 100%, and China registered a whopping 450% increase from 1996 to 2009. By mid-1990s, all these countries were among the leading producers of scientific knowledge measured through the Web of Science database. Japan was the leader, which produced 72,303 articles in 1996. Other three countries registered output ranging from 17,190 (India) to 24,125 (Australia) articles. None of the other ASEAN countries South Korea registered a mark of 10,000 articles in 1995-96. Within fifteen years, by 2009, some countries had increased their science output quite dramatically. South Korean output increased multi-fold from 8150 in 1996 to 44,330 articles in 2009; Malaysia (4951), Thailand (6253), New Zealand (8636), and Singapore (9756) also increased their science output by 2 to 4 times in this period. Even though other countries such as Indonesia (1264), the Philippines (1033) and Vietnam (1174) had registered a modest figure of above 1000 articles by 2009, each of these countries had increased their science output by almost 2 times compared to 1996.

**Table 2.** Year-wise literature growth of selected countries.

| Year | Australia | China | India | Indonesia | Japan | Malaysia | New Zealand | Philippines | Singapore | South Korea | Thailand | Vietnam | Taiwan |
|------|-----------|-------|-------|-----------|-------|----------|-------------|-------------|-----------|-------------|----------|---------|--------|
| 1990 | 16,514 | 8131 | 15,321 | 190 | 49,265 | 410 | 3576 | 298 | 943 | 1755 | 548 | 131 | 2969 |
| 1995 | 23,056 | 13,994 | 16,517 | 333 | 67,244 | 700 | 4431 | 325 | 2201 | 6600 | 760 | 228 | 7287 |
| 2000 | 27,979 | 32,289 | 18,722 | 519 | 83,889 | 957 | 5511 | 485 | 4250 | 15,341 | 1422 | 363 | 10,780 |
| 2005 | 36,175 | 76,933 | 28,206 | 708 | 91,272 | 1838 | 6901 | 713 | 7442 | 30,126 | 3112 | 658 | 17,875 |
| 2010 | 52,593 | 151,232 | 48,132 | 1318 | 93,366 | 6786 | 9285 | 1117 | 10836 | 48,198 | 6589 | 1414 | 28,131 |
| 2014 | 65,648 | 245,552 | 57,543 | 2149 | 84,541 | 10,767 | 9905 | 1279 | 12,927 | 55,695 | 6950 | 2382 | 30,674 |
| Total | 892,213 | 1,912,247 | 734,761 | 19415 | 2,022,120 | 73,710 | 162,450 | 16,356 | 150,719 | 624,545 | 76,930 | 18,631 | |

Source: Based on Krishna (2018).

## 7. National R&D, HEIs, and Human Resources

The growth of knowledge production in Asia-Pacific economies is not unrelated to increase in the R&D investments as a proportion of GDP as shown in Table 3. However, there are wide variations, and countries fall under different groupings. High-income countries (Australia, New Zealand, Japan, South Korea, Taiwan, and Singapore) have all increased their R&D/GDP investments between 2001 and 2011. Towards the end of that decade in 2011, these countries fell in the category of countries spending between (2.23%), as in the case of Singapore, and 4.03% (South Korea). The second category of countries (China, Malaysia, and India) spent between 0.81% and 1.84%. China doubled its R&D/GDP investment between 2001 and 2011. Taking into account that China's economy expanded by about three times during this period, it is emerging as the largest spender of R&D in Asia, surpassing Japan in absolute terms. The third category of countries (the Philippines, Indonesia, Thailand, and Vietnam) spent between (0.08%), as in the case of Indonesia, and 0.5% (Vietnam). All these four countries are lagging behind their Asian neighbors both in terms of knowledge production and R&D investments.

**Table 3.** Gross expenditure on R&D as a percentage of GDP.

| Countries | R&D/GDP % 2001 | R&D/GDP % 2011 | R&D/GDP % 2015–2018 |
|---|---|---|---|
| Australia | 1.5 | 2.1 | 1.92 |
| New Zealand | 1.0 | 1.1 | 1.26 |
| Taiwan | **na** | **na** | 2.4 |
| China | 0.95 | 1.84 | 2.2 |
| India | 0.73 | 0.81 | 0.62 |
| Japan | 3.07 | 3.39 | 3.5 |
| South Korea | 2.47 | 4.03 | 4.26 |
| Singapore | 2.06 | 2.23 | 2.6 |
| Malaysia | 0.65 | 1.07 | 1.3 |
| Thailand | 0.26 | 0.21 | 0.78 |
| Vietnam | 0.19 | 0.5 | 0.44 |
| Indonesia | 0.05 | 0.08 | 0.08 |
| Philippines | 0.14 | 0.11 | 0.14 |

Source: Krishna (2018); and OECD Indicators, Paris.

As shown in Table 1, with the exception of India and China, leading countries such as Japan and South Korea spent between 10% and 29% of Gross Expenditure on Research and Development (GERD) in the HEIs sector [43]. Even though both these large economies are spending less than other leading countries in the region, in absolute terms, the amount of money spent on higher education R&D witnessed more than a fourfold increase between 2001 and 2011. This is due to the relatively higher GDP growth rate registered by China and India in this decade. The proportion of total full-time equivalents (FTEs) of scientists and engineers also reveals the growing importance of the HEIs sector in 2011. A total of 80% in Malaysia, 43% in Singapore, 54% in Thailand, 32% each in Vietnam and the Philippines, and 30% in Indonesia are engaged in the HEIs sector. A total of 19% FTEs each in Japan and China, 14% in South Korea, and 11% in India are in the HEIs sector. The overall picture that is emerging is that most of the leading Asian countries have made relentless efforts in the last decade to strengthen their human capital through the promotion of Gross Enrolment Ratio (GER) in HEIs and research and innovation capacities.

## 8. Policies to Promote Innovation in Universities

Ultimately, specific science, technology, and innovation policies and institutional measures in HEIs ushered in the third mission from the 1990s with a thrust on innovation. Universities and HEIs, which were mainly seen as the most important source for highly skilled human resources and repositories of new research findings by public and private enterprises, came to also be seen as 'new frontiers of innovation'. Every Asia-Pacific country embraced and introduced policies relating to innovation in varying forms. Consultancy and collaborative links with industry being traditional forms of engagement, new policy, and institutional measures in technology transfer and innovation to engage with society and business enterprises are gaining prominence. Policies for incubation, start-ups, and spin-offs, technology transfer offices (TTOs), and science and technology parks have gained tremendous prominence in the leading Asia-Pacific universities. Two most popular concepts in higher education policies are the 'triple helix' of university-government-industry relationships and university-industry linkages (UIL). However, only a small proportion of total universities in any given country were able to effectively implement these policies, often invoking terms such as entrepreneurial or innovation universities. Some major measures and instruments in higher education policies in Asia-Pacific countries are shown in Table 4. Before the early 1990s, universities and higher education in the Asian region were the main source of skills and human resources for industry. Universities played

only a marginal role in technology transfer or innovation. Since the late 1990s, universities have begun to play a robust role in innovation. Let us examine some emerging innovation landscapes in Asia.

**Table 4.** Major policies promoting innovation in Universities.

| Countries | Major Policy Measures to Promote Innovation |
|---|---|
| Australia | Cooperative Research Centres (CRCs), 2002; National Innovation Agenda, 2016 |
| New Zealand | University Commercialization Offices, 2005 |
| Japan | Law promoting technology from universities to industry 1998; Japanese version of Bayh-Dole law was enacted and in 2001 (Hiranuma Plan); National University Corporations, 2004 |
| South Korea | Brain Korea 21 in 1999; Korean version of Bayh-Dole for TTOs, 2000; World Class University Programme, 2003; Technology Transfer and Commercialisation Promotion Act, 2006 |
| Taiwan | Fundamental S&T Act, 1999 (for innovation) |
| China | Scheme to build science parks, 1980s; 973 National Basic Research Programme, 1997; 211 and 985 Programmes (1990s); Invigorating Education Towards the 21st Century |
| India | Software Technology Parks; Science Parks; Incubation and Technology Transfer in IITs; Renovation and Rejuvenation of Higher Education, 2009 |
| Malaysia | Human Resource Development Fund, 1992; Malaysia Industry—Government, High Technology Program (MIGHT) 1993; National Higher Education Action Plan, 2011 |
| Thailand | National Science and Technology Innovation Plan, 2012 |
| Vietnam | National Fund for Technology Transfer |
| Philippines | Education Sector Analytical and Capacity Development Partnership |
| Indonesia | Ministry of Research Technology and Higher Education (UIL program); Education Sector Analytical and Capacity Development Partnership |

## 9. Emerging University Centered Innovation Landscapes

Beyond university-industry relations based on a linear model of innovation and role of universities as skills supplier, there are now various innovation landscapes emerging in some leading Asia-Pacific countries. These innovation landscapes involving universities have emerged in the Asia-Pacific countries in the last two decades. These are innovation districts, entrepreneurial universities, science and innovation parks and science, technology and innovation policies. In the next section, we will analyze the significance of these innovation landscapes spread over the region.

## 10. Innovation Districts

For the past fifty years, the most dynamic exemplar of innovation has been dominated by Silicon Valley. As is quite well known, Silicon Valley is the unique innovation landscape which has not been replicated anywhere in the world. What we have instead are different clones or versions of this geographical wonder which can be identified as innovation districts. To broadly define these, they are 'geographic areas where leading-edge anchor institutions and companies cluster and connect with start-ups, business incubators, and accelerators. Compact, transit-accessible, and technically-wired, innovation districts foster open collaboration, grow talent, and offer mixed-used housing, office, and retail' [44]. Based on the insights of Katz and Wagener, one can find a complementary new urban model of innovation districts emerging or functioning quite efficiently in various other parts of the Asia-Pacific region. We will briefly investigate three main cases, namely, Bangalore, Melbourne, and Beijing-based innovation districts.

The first case study is Bangalore as an innovation district mainly known for production, consumption, and export of ICT software. India's aerospace and biotechnology research centers are also concentrated in this city. The city has eight universities and 20 public research laboratories.

Bangalore has emerged as the fourth-largest technology cluster in the world, and the State of Karnataka is home to over 200 engineering colleges and has over 239 foreign Fortune 500 global R&D centers and research groups concentrated in the Bangalore city region [45]. Bangalore has attracted the attention of scholars around the world for its impressive software growth export rates, superior to those of competing IT hubs such as Ireland, Israel, Brazil or China. Several global Multinational Corporations (MNCs) such as Microsoft, Google, Motorola, Nortel, Intel, IBM etc. have contributed towards Bangalore's R&D landscape. A case in point is General Electric, whose presence in India began in 1902 when the company installed India's first hydropower plant in Mysore, Karnataka. John Welch Technology Centre (JFWTC) in Bangalore is GE's largest integrated multidisciplinary R&D Center and the first to be located outside the US. Over 5300 scientists and engineers work at the R&D center here and have filed more than 2250 patents [46]. India in the decade 2005–2015 generated ICT software and services revenue of US $120 to 150 billion on average annually, and more than 30% comes from the Bangalore software cluster. In 2017, this sector yielded US $154 billion [47]. More than 10% of 3 million Indian software professionals are concentrated in Bangalore and its neighborhoods.

Beyond foreign MNCs, there are several leading Indian software firms that are located in Bangalore, such as Infosys, Wipro, HCL Technologies, Tech Mahindra, and Tata Consultancy Services. R&D and innovation activity in Bangalore has quite dramatically moved up in the last decade, and half from the 'one way technology transfer' of the 1990s to 'two way and global innovation' [48]. For instance, Infosys, headquartered in Bangalore, has more than 75,000 professionals working worldwide. The importance of Bangalore as an innovation District comes into prominence due to the city's proximity to knowledge institutions, R&D centers, India's Aerospace industry, start-up ecosystem, and the ease of doing business. As Basant points out [49], the policy regime both from the state and center has aided the growth of the Bangalore innovation cluster [50]. The diaspora linkages, first mover advantage of attracting global MNC firms, decent quality of engineering, and higher educational opportunities and supply of skills, among several other factors, led to the development of this innovation district with networks of cooperation for innovation. Further, as Jan Vang draws attention [51], building the regional innovation system with a network of linkages between different actors of innovation and interactive learning within Bangalore and the national context has led to the dynamism of Bangalore.

The second case study is Melbourne's biomedical precinct and Carlton Connect precinct. These are the two most successful cases of Australia's innovation precincts or innovation clusters that have been surrounded by the University of Melbourne and other higher educational institutions. Melbourne over the years has grown as a medical and knowledge cluster of companies, start-ups, and innovation spaces. Melbourne has one of the world's largest life science clusters and is home to more than 40 per cent of Australia's biomedical researchers. The Melbourne Biomedical Precinct Office was established by the Victorian state government in late 2016 to drive economic development in the precinct and strengthen its position as a world leader in biomedical research, development, and innovation [52]. Melbourne's biomedical cluster houses more than 10,000 scientists, clinicians, and technical staff all working in proximity within a radius of two kilometers. It boasts 30 medical research institutions, hospitals, and research centers, together with over 220 pharmaceutical firms. Precinct partners employ 49,000 people, produce 7000 biomedical students, and contribute to nearly AU $3.6 billion to State's gross regional product. Spin-offs from this precinct include a AU $560 million deal for drugs to alleviate conditions relating to fibrosis and an AU $198 m deal for an innovative head lice product. In the last couple of years, Melbourne biomedical precincts attracted AU $14 billion and boasted the creation of 10,000 new jobs [53].

The significance of Melbourne is that this city is well known as the most livable city in the world that is placed within the top six cities. The collaborative spirit, particularly with the two clusters around Melbourne, Monash, Deakin, La Trobe, and Swinburne universities and the Alfred Medical Research and Education Precinct, ensures the biomedical sector is well networked with a variety of actors and agencies. These are entrepreneurs and educational institutions, start-ups and schools, mixed-use development and medical innovations, bike-sharing, the world's leading restaurants, and

sporting centers (conducting Australian Open in Tennis at the Rod Lever Arena and hosting leading Cricket tournaments at Melbourne Cricket Ground), all connected by transit, powered by clean energy, wired for digital technology, and fueled by coffee shops. Melbourne truly reflects what Saskia Sassen calls 'cityness' combined with the innovation ecosystem that has evolved over the last fifty years.

The third case study that is considered here is Beijing's Zhongguancun [54], which particularly has Peking and Tsinghua universities at its proximity. Beijing together with Bangalore and Singapore is among the top 20 tech cities in the world [55]. It is home to nearly 9000 high tech firms, including some of China's biggest internet firms, such as Nasdaq-listed Baidu and Sina. The Haidian park of Zhongguancun is home to more than 40 universities, including the world-class Peking and Tsinghua Universities, as well as more than 200 research institutes and national-level laboratories. Founded 30 years ago with a mission to "learn from Silicon Valley and replicate Silicon Valley", Zhongguancun is at the forefront of Beijing's drive to turn the country from 'workshop of the world' into a global technology powerhouse. Nearly half of China's 70 unicorns—start-ups with a valuation of more than US $1 billion each—are located in the area. According to a report, as many as 80 tech start-ups are born there every day [56]. On top of the 2018 unicorn list is ByteDance, parent of news aggregator Jinri Toutiao and video sensation Tik Tok, which was recently crowned as the world's biggest start-up with a valuation of US $75 billion [56]. Richard Liu Qiangdong chose Zhongguancun to set up a small retail shop that has now grown into China's second largest e-commerce platform—JD.com. Over the past two decades, more Zhongguancun-born companies have become household names in China by providing services from online search (Baidu) to app-enabled food delivery (Meituan Dianping). Chinese personal computer maker Lenovo Group was among the earliest tech companies to get established in Zhongguancun. After gaining the blessing of China's State Council in 1988, Zhongguancun became the country's first hi-tech industry pilot zone. Since then, the area has become the launch pad for some of China's most successful entrepreneurial firms with the backing of Beijing to make it "a national innovation centre with global influence" [57].

## 11. Entrepreneurial Universities

This is another type of innovation landscape that has been made popular by Henry Etzkowitz. He defines 'the entrepreneurial university has the ability to generate a focused strategic direction, both in formulating academic goals and in translating knowledge produced within the university into economic and social utility' [58]. As is well known, Etzkowitz has drawn this concept of entrepreneurial university from the MIT. As Phokam Wang and Singh draw attention to, 'universities around the world are increasingly shifting from their traditional primary role as educational providers and scientific knowledge creators to a more complex "entrepreneurial" university model that incorporates the additional role of the commercialization of knowledge and active contribution to the development of private enterprises in the local and regional economy' [59].

More than anything else, state mediation plays an important role for shortlisting universities to develop a strong base for research intensity. Once this is accomplished, both the university management and the government policy measures need to complement each other to translate research potential into innovation potential. Generally, this is accomplished through a series of policy measures, but one of the prerequisites is the development of an appropriate research and innovation ecosystem. In the Asian region, some of the glaring examples for the rise of entrepreneurial universities can be seen in the case of the National University of Singapore, Tsinghua University, Zhejiang University, Seol National University, and the University of Tokyo. With the exception of Japan, the concept of entrepreneurial university emerged in 2000. A systematic investigation of what constitutes an entrepreneurial university leads us to incubators, technology licensing offices, start-up culture, university science and innovation parks, and most importantly, the incentive or reward structure that places equal emphasis on publishing in high-impact journals as well as patenting leading to innovation. In some cases, government and university leadership play an important role, as in the case of National University of Singapore.

In other cases, it is the government policies that trigger and activate universities to adopt the path of entrepreneurship and innovation, as in the case of the Tsinghua and Peking universities.

## 12. Science and Innovation Parks

As part of the third academic revolution, science, and innovation parks within close proximity to universities present us with an important innovation landscape. In Europe, the publication in 1985 of *The Cambridge Phenomenon: The Growth of High Technology Industry in a University Town* by Segal Quince Wicksteed triggered the importance of science and innovation parks in Europe. Much of this story related to the success of how science parks around universities ignite innovation. The Cambridge Science Park, founded by Trinity College in 1970, is the oldest science park in the United Kingdom. It is a concentration of science- and technology-related businesses and has strong links with the nearby University of Cambridge. Later, this led to the well-known St John's Innovation Centre (SJIC) in a 21-acre area in 1987. Subsequently, the Cambridge Technopole Group was established to promote innovation and convert Cambridge's research potential to commercialization. The SJIC houses about 85 companies employing over 400 people. The Cambridge Science Park employs 6500 and houses about 104 companies. In 2018, the park alone was reported to have contributed 2.4 billion pounds to the UK's economy [60]. Similarly, there are two science parks connected with Oxford University: Oxford Science Park, which is owned by Magdalen College, and Begbroke Science Park which is owned by the University. It was estimated that the University's science parks contributed £135 million and 2382 jobs in Oxford City; £155.4 million and 2762 jobs in Oxfordshire; and £166.7 million and 3043 jobs in the UK [61]. Much of the success of these science parks is also due to the proximity of Oxbridge universities.

Given the success of science parks in Europe, Asia's leading economies begun to give high priority to building science parks in proximity to universities. We have the glaring examples of Tsukuba science city, the National University of Singapore and Biopolis, Daedeok Innopolis (formerly known as Daedeok Science Town), innovation hubs near the Tsinghua and Peking Universities, Beijing, and the science park of IIT Madras and Hsinchu Science Park of Taiwan. Among these, let us briefly look at the case of Hsinchu Science Park, which emerged in the 1980s in close proximity to two universities, namely, National Tsing Hua University (NTHU) and National Chiao Tung University (NCTU). The National Space Organization, the Industry and Technology Research Institute (ITRI), and other public research institutions are located within the close radius of Hsinchu Science Park, which is now spread over 653 hectares of land area. The significance of the innovation complex involving Hsinchu Science Park (HSP) and two universities led Taiwan to register as the world's leading country in the field of semiconductor technology from design, fabrication, packaging, and sales since the 1990s [34]. HSP houses some 570 firms mainly in the semiconductor-related technologies but is currently extended to biosciences, ICT, and other new technologies. Between 1991 and 2013, for instance, the total number of firms related to semiconductor technology (semiconductors, PC&Peripheral, communications, optoelectronics, precision machinery, etc.) increased from around 120 to 480 firms. During the same period, the sales of semiconductor-based products increased from NT $5000 (million) to NT $800,000 (million). The semiconductors sector played a significant economic role for Taiwan and as the main knowledge hub since the mid-1990s. At the end of 2016, HSP employed 147,624 persons and managed revenues over NT $1 trillion in exports. The Biomedical Park was established at HSP in 2012, which attracted 40 firms and an investment of NT $15 billion. By 2019, HSP had emerged as one of the world's leading knowledge hubs in high technology and new technologies of the 4th Industrial Revolution, such as Internet of Things and smart technologies [62]. In the science park, the applied and development research is organized in collaboration with ITRI through overwhelming focus on semiconductor devices, and the relevant basic research and skilled manpower are supported by the academic sector of two universities (NTHU and NCTU). The university, industry, and government linkages in the Hsinchu innovation region have been rapidly growing since the 2000s through the establishment of technology licensing offices and incubation centers in the NTHU, the NCTU, and

the ITRI [34]. Hsinchu Science Park in the 21st Century has emerged as one of the model science parks in Asia. HSP had demonstrated several exemplary pathways to other countries in the region for developing knowledge hubs, leading to technological leadership in some niche technology sectors.

## 13. Science, Technology, and Innovation Policies

Science, technology, and innovation policies (STI) play a very important part in not only in establishing technological capacities in a country but through certain innovation strategies, and policy measures bring about a comparative advantage to various sectors of economy. There is enough evidence to suggest East Asia and dragon economies have used STI since the 1980s to draw comparative advantage in some niche sectors. Semiconductors in Taiwan, electronics in South Korea, ICT software in India, and biomedical sciences in Singapore are some of the examples. In this sense, STI can be considered as one of the important segments of innovation landscapes. Chinese STI in the higher education and universities sector is the most glaring example that deserves special attention. In the 1980s and 1990s, none of the Chinese universities figured in the list of world's leading top 100 universities. In little more than two decades since the 1990s, four universities were listed in the Quacquarelli Symonds World University Rankings, 2017 in the top 100; and another three in the top 200 list. More than half a dozen Chinese universities currently play a significant role as major collaborators in boosting public sector state enterprises (High Speed Rail Network and commercial aircraft industry, etc.) and private sector enterprises (Alibaba group, Huawei, Lenovo, etc.) of the Chinese economy. In a significant way, Chinese universities have become frontiers of innovation and skills in a large measure. Among various factors, STI in higher education and universities stands out as one of the important actors or a pathway in innovation landscapes. From this perspective, let us briefly look into important milestones of STI higher education policy regime [63].

In 1995, the central government launched "Project 211" with the intent of raising the research standards of around 100 universities. The implementation of "Project 211" is an important measure taken by the Chinese government in its effort to facilitate the development of higher education in the context of the country's advancement in social and economic fields. Primarily aiming at training high-level professional manpower to implement the national strategy for social and economic development, Project 211 was an important reform measure for improving higher education, research intensity in universities and promotion of science, technology, and innovation for international competitiveness. Project 211 ran in three phases with substantial funding. The first phase (1995–2000) covered 99 universities and 602 key disciplines with a funding of 10.32 billion Yuan. The second phase (2001–2005) covered 107 universities and 821 (to be counted cumulatively), with a funding of 6.78 billion Yuan. The third phase ran from 2008 to 2011, covering 112 university and 1073 key disciplines with a funding of 10 billion Yuan. In principle, projects were aimed at strengthening key disciplines in selected universities to enhance their research intensity. With the continuation of this project, another reform measure was introduced in the late 1990s known as Project 985.

Project 985 was first announced by CPC General Secretary and Chinese President Jiang Zemin at the 100th anniversary of Peking University on May 4, 1998. One of the main objectives was to promote the development and reputation of the Chinese higher education system towards attaining the status of world-class universities. The project involved both national and local governments allocating large amounts of funding to certain universities in order to build new research centers, improve facilities, hold international conferences, attract world-renowned faculty and visiting scholars, and help Chinese faculty to attend conferences abroad. In 1998, 10 of China's Project 985 universities were allocated budgets for three years in excess of 30 billion RMB for quality improvements. Some the prominent universities in this phase included Peking, Tsinghua, Fudan, Zhejiang, and Nanjing universities. The second phase of Project 985 began in 2004 and ended in 2007, with 39 universities distributed in 18 provinces and municipalities. The top 11 universities (including the first phase universities) were allocated 17.43 billion RMB. In 2009, the original nine founding member universities of Project 985

formed the C9 League, which is referred to as the Chinese equivalent of the US Ivy League. By the end of the second phase of the project, 39 universities were sponsored.

Another important project that was introduced was the 973 Program, which had the goal of inducing networking between scientific research institutions, universities, and enterprises. The 973 Program produced a large number of PhDs and Master's degrees. In 1998, the number of postdoctoral workstations was 1897, and in 2007, the number of postdoctoral stations had reached 7903, with a postdoctoral scale of about 10,000. The development of postdoctoral research in many ways contributed to the science output of China from the late 1990s. China's total Science and Technology (S&T) output of research papers increased from 8131 in 1990 to 32,289 in 2000; and further increased to a whopping figure of 175,329 by 2011 when the Project 211 came to close [41]. The universities accounted for more than 80% of these publications.

Higher education policies since the mid-1990s, particularly the Project 211 and Project 985, strengthened the research intensity and innovation ecosystem of a select group of universities. More than a dozen universities have developed considerable research and intellectual capacities to compete at the global level and at the same time contributed to national socioeconomic development in recent years. In ARWU (2017) rankings, 45 universities are in the list of global top 500 universities; 9 universities are in the list of global top 200; and 2 universities in the world's top 100 universities. In the TIMES (2017) rankings, 6 universities entered the world's top 200; and 2 entered the world's top 100 universities. In QS (2017) rankings, 15 universities entered the global top 400; 7 entered the global top 200 universities; and 4 universities entered the global top 100 universities. In USNEWS, 2017 rankings, 30 universities were in the world's top 500; 7 ranked in the world's top 200; and 2 universities were listed in the world's top 100. The rise of China in global knowledge production and technological innovation is closely related to the rise of a dozen Chinese universities in research and innovation capacities.

## 14. Concluding Remarks

Historically speaking, universities in the Asia-Pacific region have gone through three academic revolutions or phases of transformation in the post-war era. The three academic revolutions that were identified and explored in this paper have led to three institutional missions in the role of universities, namely, teaching, research, and innovation. Even though there are significant variations in the way in which all these three missions are combined in the role played by universities, there emerged three types of universities, namely, teaching universities; teaching and research-based universities; and the universities which combine all the three missions of teaching-research-innovation. Across the Asia-Pacific region, one can clearly identify the role of universities and higher educational institutions emerging as important actors in the NIS to impact both economy and society. The NIS perspective was useful to understand how different university and higher educational systems were structured and organized to perform different roles in the national systems. More than anything else, this perspective was useful to differentiate various institutional arrangements and policies in a comparative perspective to examine how various elements and factors either facilitated and promoted or even constrained the mission of universities in a particular pathway.

The orientation of teaching institutions to impart skills and training to emerging service and new industrial sectors is a major shift that has come about in the last decade and a half in the Asia–Pacific region. Together with the massification of higher education and remarkable increase in GER in the Asian region, the share of the private sector's role in higher education has witnessed tremendous growth. Enrolments in private higher education as a percentage of total higher education registered figures of 81% for Japan; 79% for South Korea; 64% for Singapore; 63% for the Philippines; and 62% for Indonesia.

The NIS perspective yielded three groupings. In the first group are Australia, Japan, South Korea, China, Taiwan, and Singapore, which are investing between 2% to 4% of their GDP in R&D. All these countries not only enacted robust UIL and triple helix policies but have also made substantial investments in making specific institutional arrangements to foster and promote them. In the second

group are India, Malaysia, and New Zealand, which are investing from 0.75% to 1.25% of their GDP in R&D and which have made only modest institutional arrangements and investments to promote UIL and triple helix measures. In the third group, we have SE Asian economies (Indonesia, the Philippines, Vietnam, and Thailand), where one can find the real gap between policy rhetoric on UIL, triple helix, and their implementation. Much of the inability of these countries to make institutional arrangements and implement policies to foster UIL and triple helix seems to be due to very low level of their national investments in R&D.

The NIS perspective was useful in exploring certain historical developments in the Asia-Pacific region. Japan and South Korea, in their initial stages of industrialization from the1970s to 1990s, were heavily dependent on highly skilled educated work force trained in specialized technical and higher educational institutions. Institutional arrangements were organized in such a way as to facilitate large business corporations (Sony, Mitsubishi, Toyota, Hyundai, and Samsung, for example) to exploit technological frontiers in achieving international competitiveness. Much of the technological advancement and generation of wealth in Japan, South Korea, and Taiwan came about through knowledge and technology institutions, thereby reflecting more 'advanced environments' compared to SE Asian countries. Historically speaking, unlike Australia and New Zealand, these three Asian economies advanced technologically through science, technology, and higher educational institutional base.

Australia and New Zealand, unlike Japan, South Korea, Taiwan or Singapore, are not relatively driven and dependent on scientific and technological factors feeding into national growth or GDP. Historically speaking, this background has generated much less pressure on universities. Universities and higher education in both these countries have achieved relatively high international reputations in teaching and research. They are held in very high esteem by neighboring East, South East, and South Asian countries. Exporting education and generating revenues have become important factors in Australian and New Zealand's higher educational system. As prospects of economic boom through the resource base and primary industry began to fade away, a new policy focus on innovation emerged in Australia from the beginning of the 21st century.

In China, from the late 1980s, a series of public policies directed at higher education and universities induced research intensity and innovation cultures in a select group of leading Chinese universities. The 973 National Basic Research Program in 1997, Project 211, and Project 985 allocated large amounts of research and innovation funding in select universities to develop them into world class universities. By 2019, five Chinese universities had figured in the world's top 100 universities and induced strong innovation cultures aiding large firms, such as Alibaba and Huawei. Unlike China, India did not become proactive in promoting UIR or initiating specific policies with adequate R&D investments in the university sector. Six Indian Institutes of Technology are the leading universities with links in the software sector. The major policy push by the government was in the creation of software technology parks since the 1990s in a few major cities such as Bangalore, Hyderabad, Delhi, Chennai, and Pune and Hyderabad.

A major transformation that has come about in the last decade in universities is the diversity of innovation activities that have been institutionalized. These activities include establishment of incubation and start-up mechanisms, technology transfer offices, innovation and entrepreneurship centers, science and technology parks, and specific centers that promote university spin-offs on the campuses. The two most prominent concepts institutionalized by universities are the 'triple helix' of university-government-industry relationships and university-industry linkages (UIL). Only a small proportion of total universities in any given country represent these types of universities, which are often called entrepreneurial universities or innovation universities. The increasing thrust given to higher education in the Asia-Pacific region is intimately associated with innovation activities in universities, particularly, the establishment of incubation and start-up mechanisms, technology transfer offices, innovation and entrepreneurship centers, and science and technology parks. Different countries have adopted different mechanisms and strategies in locating and positioning universities in the

emerging innovation landscapes. One can identify five innovation landscapes involving universities in the Asia-Pacific region. The first and traditional landscape is the role of universities as the main supplier of human skills and educated human resources. In the other four innovation landscapes, universities play a robust role in technology transfer and innovation. These are innovation districts, entrepreneurial universities, science and innovation parks, and science, technology, and innovation policies.

**Author Contributions:** The whole paper is structured and written by V.V.K. who carried out research in Asia-Pacific countries in the last four years. Much of the field research was undertaken during several trips to various Asian countries, Australia, and New Zealand.

**Funding:** This paper was presented as a keynote speech of SOItmC 2019, and the publishing fee was supported by SOItmC.

**Acknowledgments:** I would like to thank following colleagues: Ching-Yan Wu, Mei-Chih Hu, Rajiv Mishra, Swapan Patra, Joseph Yun, and others.

**Conflicts of Interest:** Some material and portions of the paper were drawn from the author's own research and earlier research papers and books. These are fully acknowledged and there is no conflict of interest in this regard.

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
