# Peer review of "Universities in the National Innovation Systems: Emerging Innovation Landscapes in Asia-Pacific"

_2199-8531, doi:10.3390/joitmc5030043_

Reviewer 1 Report

The manuscript explores the evolution of Higher Education Institutions (HEIs) in Asia- Pacific, and their contribution to National Innovation Systems in the respective countries. It describes thoroughly the three stages in the evolution of University (teaching institution, teaching + research and focus of innovation) with some examples from Asia Pacific Economies.

In fact, the region harbours the fastest developing economies at world level (among them China, India, Japan), which might be related to the high investment in education and innovation, the subsequent high enrolment rate in Higher Education (much higher than in Europe) and the institutional support to innovation and entrepreneurship, in the form of innovation landscapes (e.g. innovation parks or innovation districts).

The paper presents solid evidence that proves the close relationship between investment in education, scientific production, entrepreneurship and economic development and, in this sense, results inspiring and encouraging for those economies wanting to enhance their R&D sector.

As for the text, it is extremely dense, with a lot of information, that is often difficult to follow, as the discourse progress simultaneously in two axes (chronological first, then thematic + several countries). It might be helpful to use some graphical aids to help interpreting the main points.

Likewise, the weakest part of the otherwise quite nice paper is the conclusions section. It’s just a recollection of some of the findings already presented in the text. However, I feel the authors loose there their opportunity to make some strong points and help the reader interpret much more easily what can be learned from the cases examined.

Another question is whether it should be considered a research article, or rather an essay, as there is not investigation involved.

Author Response

The manuscript explores the evolution of Higher Education Institutions (HEIs) in Asia- Pacific, and their contribution to National Innovation Systems in the respective countries. It describes thoroughly the three stages in the evolution of University (teaching institution, teaching + research and focus of innovation) with some examples from Asia Pacific Economies.

OK 

In fact, the region harbours the fastest developing economies at world level (among them China, India, Japan), which might be related to the high investment in education and innovation, the subsequent high enrolment rate in Higher Education (much higher than in Europe) and the institutional support to innovation and entrepreneurship, in the form of innovation landscapes (e.g. innovation parks or innovation districts).

OK

The paper presents solid evidence that proves the close relationship between investment in education, scientific production, entrepreneurship and economic development and, in this sense, results inspiring and encouraging for those economies wanting to enhance their R&D sector.

OK

As for the text, it is extremely dense, with a lot of information, that is often difficult to follow, as the discourse progress simultaneously in two axes (chronological first, then thematic + several countries). It might be helpful to use some graphical aids to help interpreting the main points.

 Answer: We are dealing with cross country comparison in Asia-Pacific region covering a number of countries. Hence, the material used is dense and a mix of historical, thematic and teasing out various important points.

----

Likewise, the weakest part of the otherwise quite nice paper is the conclusions section. It’s just a recollection of some of the findings already presented in the text. However, I feel the authors loose there their opportunity to make some strong points and help the reader interpret much more easily what can be learned from the cases examined.

OK will try to add some strong points.

Another question is whether it should be considered a research article, or rather an essay, as there is not investigation involved.

Of course, it is a research piece. Otherwise how can one expect insights into so many Asian countries. It is certainly NOT  AN ESSAY OR REPORT PLEASE.

Reviewer 2 Report

The paper focuses on Asia-Pacific and explores broad historical trajectories of innovation systems and STI policy implications in them. The paper could be more concise and it could present its purpose (main research question) with one or two points. Figure 1 is a copy-paste and it should be redrawn. There are also some variations in the writing (e.g. Asia-Pacific and Asia Pacific). Language check could be in order. In some sections (particularly Science, Technology and Innovation Policies) is tiresome to read due to an extensive amount of numerical figures covering a number of topics. Tables could help but some of them (1 and 2) are embedded into the text and then 3 and 4 just listed after the references. Overall, the paper is rather difficult to follow and it could improve significantly if extensively modified in terms of structure and purpose. The conclusions could also be more tight up to the main point of the paper.

Author Response

The paper focuses on Asia-Pacific and explores broad historical trajectories of innovation systems and STI policy implications in them. The paper could be more concise and it could present its purpose (main research question) with one or two points. Figure 1 is a copy-paste and it should be redrawn. There are also some variations in the writing (e.g. Asia-Pacific and Asia Pacific). Language check could be in order. In some sections (particularly Science, Technology and Innovation Policies) is tiresome to read due to an extensive amount of numerical figures covering a number of topics. Tables could help but some of them (1 and 2) are embedded into the text and then 3 and 4 just listed after the references. Overall, the paper is rather difficult to follow and it could improve significantly if extensively modified in terms of structure and purpose. The conclusions could also be more tight up to the main point of the paper.

Answers

1. Will try to improve and redraw the figure.

2. Will use Asia-Pacific through out

3.Journal could use the tables according to its style. No issues

4. Will pay attention to conclusions

Reviewer 3 Report

I think there is significance as an article focusing on Asia-Pacific Emerging Innovation.
But it was hard to read and easy to understand.
I would like to cluster the contents of 12 chapters and rearrange them according to the theme.

Author Response

Will try to pay some attention to the comments and suggestions.